# EFFICHRONIC study protocol: a non-controlled, multicentre European prospective study to measure the efficiency of a chronic disease self-management programme in socioeconomically vulnerable populations

An L D Boone [ID],[1] Marta M Pisano-Gonzalez [ID],[2] Verushka Valsecchi,[3] Siok Swan Tan [ID],[4] Yves-Marie Pers [ID],[3] Raquel Vazquez-Alvarez,[2] Delia Peñacoba-Maestre,[2] Graham Baker,[5] Alberto Pilotto,[6] Sabrina Zora,[6] Hein Raat [ID],[4] Jose Ramón Hevia-Fernandez,[1] on behalf of the EFFICHRONIC Consortium

For numbered affiliations see end of article.

**Correspondence to**
An L D Boone;
An.Boone@asturias.org

## ABSTRACT

**Introduction** More than 70% of world mortality is due to chronic conditions. Furthermore, it has been proven that social determinants have an enormous impact on both health-related behaviour and on the received attention from healthcare services. These determinants cause health inequalities. The objective of this study is to reduce the burden of chronic diseases in five European regions, hereby focusing on vulnerable populations, and to increase the sustainability of health systems by implementing a chronic disease self-management programme (CDSMP).

**Methods and analysis** 2000 people with chronic conditions or informal caregivers belonging to vulnerable populations, will be enrolled in the CDSMP in Spain, Italy, the UK, France and the Netherlands. Inclusion of patients will be based on geographical, socioeconomic and clinical stratification processes. The programme will be evaluated in terms of self-efficacy, quality of life and cost-effectiveness using a combination of validated questionnaires at baseline and 6 months from baseline.

**Ethics and dissemination** This study will follow the directives of the Helsinki Declaration and will adhere to the European Union General Data Protection Regulation. The project's activities, progress and outcomes will be disseminated via promotional materials, the use of mass media, online activities, presentations at events and scientific publications.

**Trial registration number** ISRCTN70517103; Pre-results.

## INTRODUCTION

In the 21st century, chronic conditions are among the most important public health challenges mainly due to their double impact: the human suffering they cause on the one hand and the damage they produce to global socioeconomic development on the other. Even though healthcare systems are still primarily focusing on acute diseases,[1] they are slowly transforming because of demographic changes, the increasing prevalence of chronic diseases and the budgetary crisis of recent years. This shift requires new healthcare delivery models to ensure sustainability.

Chronic conditions and also vulnerable settings[2] are part of the 10 threats to global health the WHO has warned of recently. More than 70% of mortality worldwide (41 million people each year), is due to chronic conditions, such as cancer, diabetes, respiratory disorders and heart disease. All of them are

### Strengths and limitations of this study

► Comparable data on the chronic disease self-management programme implementation in five European countries with different backgrounds will be collected.
► The outcome of the used recruitment strategies, specifically designed for the vulnerable study population, will lead to innovative conclusions.
► The study is not randomised controlled to allow all possible beneficiaries to participate.
► One of the limitations of the study is the fact that vulnerable people, who do not speak or understand the local language, cannot be included.
► Ideally, a longer-term trial should be conducted to measure the outcomes at 1 year postintervention.

related to five main risk factors: tobacco consumption, physical inactivity, alcohol abuse, unhealthy diet and air pollution. Furthermore, 22% of the global population lives in vulnerable settings which exist in almost all regions of the world. As a result, the 13th WHO General Programme of Work (2019–2023), expresses this concern with the slogan: 'Promote health, keep the world safe, serve the vulnerable', urging particularly to address health promotion in populations suffering vulnerability.[3]

The large impact of non-communicable diseases (NCDs) on national income is primarily due to loss of productivity caused by absenteeism and inability to work which threaten the sustainability of recent macroeconomic achievements. Additionally, there has been a steady global increase in healthcare expenditure on NCDs over the years. According to a systematic review on healthcare spending and national income, cardiovascular diseases account for 12% of healthcare expenditure in the European Union (EU) and diabetes-related costs were estimated at 7%.[4] Besides, other conditions and risks factors such as depression or obesity are becoming more prevalent in higher-income EU countries and are greatly determined by poverty and low income.[5 6]

Also, recent scientific evidence has shown the enormous impact of social determinants (ie, income, unemployment, housing quality, educational level or gender) on health, which manifest in diverse short-term and long-term health inequalities.[7] These differences are not only related to deficient self-care behaviour[1] and poor health but also to the insufficient attention from healthcare services. Vulnerable people are generally less likely to receive preventive care, timely diagnosis or follow-up care for their chronic conditions. These observations agree with the 'inverse care law' theory described in 1971[8] by Tudor Hart, referring to the fact that often, the availability of good medical or social care varies inversely with the need of the population served. This way, chronic conditions and deprivation create a vicious circle.

Fortunately, there is evidence that chronic conditions can be controlled by reducing their risk factors,[9] which are mostly linked to unhealthy lifestyle. There is also proof that health literacy programmes that focus on self-efficacy and empowerment can successfully modify many harmful habits.[10 11]

The basis of the EFFICHRONIC project is the internationally acknowledged chronic care model[12] which aims to promote patients' empowerment and autonomy. The specific programme used in the project is the chronic disease self-management programme (CDSMP).[13] It concerns a process-centred intervention constructed on the social learning theory[14] and it is appropriate for adults living with one or more chronic conditions. Since its beginning in 1990,[15] the effectiveness of the CDSMP has been demonstrated through positive health outcomes and more efficient use of health resources.[16] The programme is community-focused and disseminated through an international infrastructure of training, certification and licensing.

## Study rationale

Up to date, prevention research still concentrates mainly on disease-specific interventions and few studies on particularly vulnerable populations are performed. Moreover, data on the efficiency and cost–benefit of prevention programmes are scarce, which reduces policy engagement and likewise, it prevents health systems from being more sustainable and efficient.[17 18]

EFFICHRONIC aims to empower individuals from (socially) deprived communities to manage their chronic conditions, in an integrated and non-disease specific way. The overall objective of the project is to reduce the burden of chronic conditions and increase the sustainability of health systems by implementing a CDSMP in socioeconomically vulnerable populations in five European regions.

Five specific objectives have been defined: (1) to identify vulnerable people by means of a multidimensional analysis in the involved regions; (2) to design specific recruitment strategies to reach the target population; (3) to implement the programme in the five regions with at least 400 individuals per region; (4) to generate a comprehensive impact assessment framework, including cost-efficiency and health-economic assessment; and (5) to define guidelines and policy recommendations to allow to the scaling up of the EFFICHRONIC methodology to other regions in Europe.

## METHODS AND ANALYSIS
### Methodology

The methodology used for the community-based intervention of EFFICHRONIC is the 'chronic disease self-management programme' developed at Stanford University. The CDSMP is a public health intervention provided by trained monitors. The programme is strongly evidence-based and consists of the following conceptual elements: self-efficacy, empowerment, salutogenesis, peer-to-peer education and vicarious training. It includes action planning and feedback about problem-solving skills and other competencies such as the reinterpretation of symptoms and training in disease management.

A cascade training model is used in the CDSMP method. On top are Self-Management Resource Centre licensed T-trainers possessing the maximum teaching expertise. They are in charge of training master trainers, who in their turn train monitors (or leaders) who will conduct the chronic disease self-management workshops. The monitors receive practical training on the philosophical and methodological basis of the CDSMP: empowerment, self-efficacy, group management, management of painful emotions, positive health and efficient communication among other skills. Once trained, the monitors facilitate the workshops in pairs. In each country, minimum 32 monitors will be trained to instruct the end-users: people with chronic conditions or caregivers. The CDSMP workshops last for 6 weeks with sessions of 2.5 hours once a week in groups of 12–20 participants. The programme

fidelity and quality plan includes the supervision of workshops in the community by the master trainers through periodic protocolled monitor audits. Workshop assistants receive a handout with the workshop topics and homework assignments, a consultation book and a certificate of attendance if they were present at minimum 4 of the 6 sessions.

## Study design

EFFICHRONIC is a multicentre non-controlled prospective study with a duration of 3 years in five European settings: the Principality of Asturias in Spain, the area of Genoa in Italy, several regions in the UK, the Occitanic region in France and the area of Rotterdam in the Netherlands. The participating countries represent a wide range of social, economic and healthcare-related realities. The EFFICHRONIC intervention consists of the following phases: identification, recruitment and stratification of participants from vulnerable populations, implementation of the intervention and an economic and efficiency evaluation with a pre–post design. Additionally, guidelines and policy recommendations will be developed based on the project conclusions, this to allow the scaling-up of the intervention to other contexts.

## Study population

The final study population will consist of 2000 vulnerable participants distributed over the five implicated areas, who finish the CDSMP (4 out of 6 sessions) and who are evaluated 6 months after the intervention. Taking into account a drop-out rate of minimum 20%, at least 2500 participants will be recruited initially.

The vulnerability concept of EFFICHRONIC is based on a definition from the Spanish Red Cross which describes it as 'a situation between complete social inclusion and total exclusion'.[19] Keeping in mind this definition and adapting it to some specific requirements of the CDSMP (eg, being able to understand the local language) and of EFFICHRONIC (being traceable for follow-up at 6 months), the inclusion criteria were defined (figure 1).

The target population involves two main groups: vulnerable people with at least one chronic condition, and informal caregivers who are socially isolated. In general, participants must be minimum 18 years old, reside in the selected geographical areas, have their basic housing needs met and possess adequate knowledge of the local language.

Participants with chronic conditions are either people older than 65 living alone or in a nursing home; people having difficulties to make ends meet (especially if belonging to an ethnic minority or being a legal immigrant, asylum seeker or refugee); or people who are imprisoned. A chronic disease, self-reported or clinically evaluated, is outlined as a pathology listed in the International Classification of Primary Care (ICPC-2) with a code between 70 and 99 in 1 of the 17 chapters and should have >6 months of evolution.

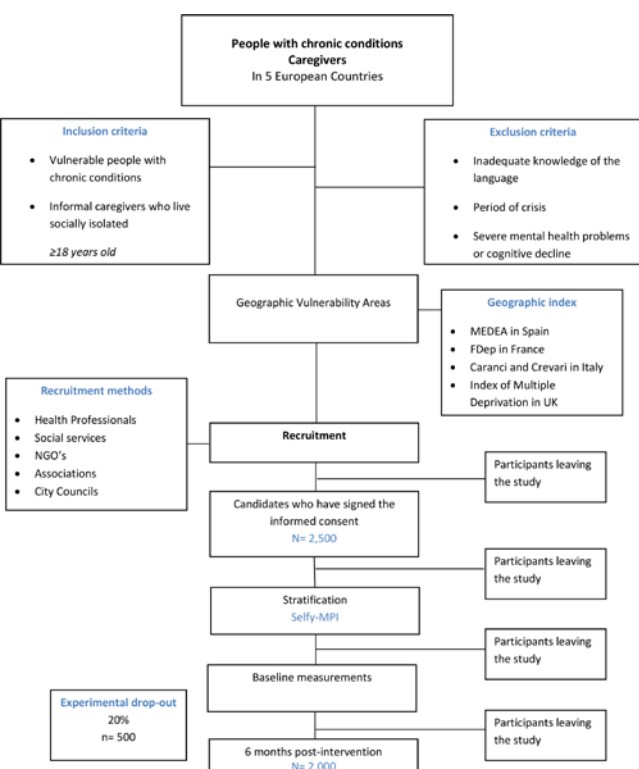

**Figure 1** EFFICHRONIC: inclusion criteria flowchart. MPI, Multidimensional Prognostic Index; NGO, non-governmental organisations.

Informal caregivers are defined as people looking after a sick person, usually a relative, without receiving payment. Additionally, to be included in the study, it is required that they are socially isolated, as is the case of carers living in remote areas without transportation or with limited access to it (without a private car and public transport at >1 km from the house), without internet access or little social support.

General exclusion criteria are: going through a period of crisis (eg, being evicted), having severe mental health problems that cause a distorted perception of reality and/or inability to participate in group dynamics, having cognitive decline (eg, Alzheimer's) or having active addictive disorders. Although the excluded groups are among the most vulnerable of society, they need other and more specific interventions before being eligible for a self-management programme.

## Sample size

The study primarily wishes to demonstrate a statistically significant difference in the two subscales of the 12 items Short Form (SF-12)[20] measuring health-related quality of life, and in the five participating countries. Based on a study on the effect of the CDSMP in lumbar back pain, the subscores 'well-being' and 'energy' of the SF-36 improved by 6.0±19.1 and 4.3±23.2, respectively, after 6 months.[21] The lowest Cohen's d is therefore of 0.185 in quality of life domains. Thus, the principal conclusion of the study will rely on 10 tests. Applying the Bonferroni's correction on the significance threshold, we will consider

an alpha risk of 0.005 for the tests measuring the quality of life (self-rated with SF-12) in vulnerable patients with chronic diseases and caregivers. To show a statistically significant effect size of 0.185, by a paired Student's t-test, with a bilateral alpha risk of 0.005, a power of 0.9 and a correlation between baseline and 6 months of 0, we need to analyse 978 subjects. Assuming a drop-out rate of 20%, we need to include at least 1223 subjects. The multicentre study will include 2000 subjects (400 per country), which will bring about sufficient power to analyse the objectives.

## Sample selection and recruitment

The potential participants are being identified by means of geographical, socioeconomic and clinical stratification processes. A distinction is made between the individual and social dimension of vulnerability, leading to different recruitment strategies that will be combined to be more efficient. First, geographical mapping of areas with a higher percentage of vulnerable population was done based on existing national and local indices in all participating countries. Consecutively, individualised recruitment will take place in these areas employing specifically designed actions. Multiple recruitment channels are used in coordination with local agents from the healthcare system, social services and community-based structures. More specifically, the following profiles and organisations are involved in recruitment: health professionals, pharmacists, social workers, non-governmental organisations, city councils and associations. The recruitment strategy at each study site is adapted to the specific local circumstances and collaborating organisations.

## Stratification

To identify the people who will benefit most from the programme, individual socioeconomic and clinical parameters will be measured by a multidimensional stratification analysis. The tool used for this purpose, the 'SELFY-MPI', has been developed based on the validated Multidimensional Prognostic Index.[22] The new tool is self-administrable and was specifically designed for EFFICHRONIC,[23] adapting it to the broad adult population (not limited to the elderly) and incorporating the socioeconomic parameters from the Spanish Gijón scale: education, income, cohabitation, age and social support.[24] Income categories are country-specific and built on the corresponding minimum wages.

## Implementation

Once the target population is identified and recruitment is ongoing, the programme can be implemented. To reach the target sample size of 2000 people, between 100 and 140 workshops (20–28 per country) will have to be conducted. In practice, for about a year and a half, one workshop will start every month at each study site.

## Evaluation

The last phase of the project consists of evaluating the effectiveness of the programme when it comes to changing health-related behaviour, self-efficacy and quality of life.

Furthermore, the economic and efficiency impact will be analysed through cost–benefit analyses. An autoadministrable impact questionnaire, containing validated scales and measuring tools, has been designed to assess these different outcome measures at baseline (T0, before the CDSMP) and 6 months (T1).[25]

## Statistical analysis plan

Differences between baseline and follow-up measurements will be assessed employing the paired T-test (for normally distributed variables), the Mann-Whitney U test (for variables not normally distributed) and the Pearson $\chi^2$ test (for variable fractions). To adjust for multiple testing, one-way analyses of variance with post hoc testing (type Bonferroni) will be performed. The relationship of an outcome measure (T1) with explanatory variables will be determined using ordinary least squares regression analysis. The outcome measure (T0), (change in) other outcome measures, risk factors and social determinants may serve as explanatory variables. Additionally, the abovementioned analyses will be repeated for each country separately and possibly for other subgroups (for variables which are likely to influence the effect of the intervention itself, eg, age and gender). All data from each of the five pilot sites will be anonymised and centralised at one study site for statistical analysis. Analyses will be performed with SPSS V.25.0.

## Guidelines and policy recommendations

By the end of the project, guidelines and policy recommendations will be formulated based on the specific results of the participating countries. These recommendations will take into account acknowledged best practices at European level, to allow the scaling up of community-based interventions in vulnerable populations.

## Patient and public involvement

Patients were not involved in the design of the study, choice of outcome measures or definition of the recruitment strategies. However, some patients and caregivers take an active part in the conduct of the study as trained monitors, facilitating workshop sessions in the community. Study participants will be able to access the study results on request as stated in the participant information letter. News on the study progress is being published via various general and specialised media channels throughout the running of the study.

## ETHICS AND DISSEMINATION

It is not considered ethical to conduct descriptive research on vulnerable subjects without running specific interventions afterwards. For this reason, the study is not randomised-controlled, being consistent with the principle of social justice and the ethics of working with marginalised groups.[26]

The study follows the directives of the Helsinki declaration and the corresponding ethical regulations are

being respected at each study site. In Occitanie, France, the Ethics Committee of the South-west and Oversees I in Toulouse (Comité de Protection des Personnes Sud-Ouest et Outre-Mer I) approved the study on 05 November 2018; study number 9788. In Genoa, Italy, the Regional Ethics Committee of Liguria (Il Comitato Etico della Regione Liguria) approved the study on 27 March 2018; study number 152–2018. In Rotterdam, the Netherlands, the Medical Ethics Review Committee (Medische Ethische Toetsings Commissie; METC) of the Erasmus MC University Medical Center—Rotterdam approved the study on 23 November 2017; study number MEC-2017–1116.

In Asturias, Spain, the Research Ethics Committee of the Principality of Asturias (Comité de Ética de la Investigación del Principado de Asturias) approved the study on 31 January 2017; study number 20/17. In the UK, the online decision tool and query line of the Health Research Authority were consulted and it was concluded that approval from the NHS Research Ethics Committee was not necessary.

Eligible participants will be informed of the objectives and the procedures of the CDSMP in person by the investigator and in writing by a participant information letter before enrolment. Only after a sufficient period of reflection and after clarification of all questions, the participant will be asked to sign two copies of the informed consent form. Participation in the study is completely voluntary and participants have the right to withdraw their consent from the study at any time for any reason.

Data processing, communication and transfer are done following the EU Regulation 2016/679 of the European Parliament and of the Council of 27 April 2016, also known as the General Data Protection Regulation.[27] Only the research team that has to maintain confidentiality has access to all collected study data. Only impersonalised information may be transmitted to third parties. Information transmission to other countries will be carried out anonymously with a data protection level according to the mentioned regulation. The data will be collected and preserved in a codified way until the study is finished. The person responsible for data custody is responsible for the data file and treatment. At the end of the study, the data will be anonymised.

Via promotional materials, use of mass media, online activities, presentations at events and scientific publications, the project's activities, progress and outcomes will be disseminated as described in a specific communication and dissemination plan. The strategy aims to enhance the engagement of healthcare and social care providers, civil society and public administration officers, all of which multipliers on account of their broad network and potential influence on the target population. Nevertheless, the main intended beneficiaries of the strategy are vulnerable people with long-term conditions and the health services serving these people. Accessing the hard-to-reach is one of the major challenges in EFFICHRONIC for which a solid strategy will be used.

**Author affiliations**
¹Public Health General Directorate, Ministry of Health of the Principality of Asturias, Oviedo, Spain
²Primary Health Care, Health Service of the Principality of Asturias, Oviedo, Spain
³Immunologie Clinique et Thérapeutique Ostéo-articulaire, Centre Hospitalier Universitaire de Montpellier, Montpellier, Languedoc-Roussillon, France
⁴Public Health, Erasmus Medical Center, Rotterdam, The Netherlands
⁵QISMET, Portsmouth, UK
⁶Geriatric Care, Ente Ospedaliero Ospedali Galliera, Genova, Liguria, Italy

**Acknowledgements** The authors would like to thank reviewer Mr Cesar Fernandez Lazaro for his critical reading and valuable suggestions for improvement, Sabine Talloen for her thorough revision and language editing, and Rosa Carretero de Lama for her administrative support.

**Collaborators** The members of the EFFICHRONIC Consortium are: Marta M Pisano Gonzalez, Raquel Vazquez Alvarez, Delia Peñacoba Maestre (Health Service of the Principality of Asturias – SESPA, Spain); An L. D. Boone, José Ramón Hevia Fernandez and Sergio Valles García (Ministry of Health of the Principality of Asturias – CSPA, Spain); Sonia López-Villar, Inés Rey Hidalgo and Raquel Ochoa Gonzalez (the Foundation for the Promotion in Asturias of applied Scientific Research and Technology – FICYT, Spain); Yves-Marie Pers, Christian Jorgensen, Verushka Valsecchi, Rosanna Ferreira, Adrien Durand, Cristina Balaguer Fernandez, Dallal Fracso (The University Hospital Center of Montpellier – CHUM, France); Graham Baker, Danni Brown and Suzanne Lucas (Qismet, Portsmouth, UK); Siok-Swan Tan, Irene Fierloos, Xuxi Zhang, Petra de Vries, Hein Raat (Erasmus MC University Medical Center, Rotterdam, The Netherlands); Alberto Pilotto, Sabrina Zora, Alberto Ferri, Alberto Cella and Alessandra Argusti (EO Galliera Hospital, Genua, Italy); Ascensión Doñate Martínez, Laura Llop Medina and Jorge Garcés (University of Valencia, Polibienestar Research Institute – UVEG, Valencia, Spain).

**Contributors** MMP-G designed the initial study protocol. ALDB, VV, SST, Y-MP, RV-A, DP-M, GB, AP, SZ, HR and JRH-F advised on the study protocol. ALDB and MMP-G drafted the manuscript. All authors critically revised the manuscript on important content and approved the final manuscript.

**Funding** This work was supported by the European Union's Health Programme (2014-2020) grant number 738127.

**Competing interests** None declared.

**Patient consent for publication** Not required.

**Provenance and peer review** Not commissioned; externally peer reviewed.

**ORCID iDs**
An L D Boone http://orcid.org/0000-0001-5198-8889
Marta M Pisano-Gonzalez http://orcid.org/0000-0002-6485-9370
Siok Swan Tan http://orcid.org/0000-0002-9522-2321
Yves-Marie Pers http://orcid.org/0000-0001-5927-3773
Hein Raat http://orcid.org/0000-0002-6000-7445

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
