## [Reviewer comments · BMJ Open]

ARTICLE DETAILS

TITLE (PROVISIONAL)	EFFICHRONIC study protocol: a non-controlled, multicentre European prospective study to measure the efficiency of a Chronic Disease Self-Management Programme in socio-economically vulnerable populations.
AUTHORS	Boone, An LD; Pisano-Gonzalez, Marta; Valsecchi, Verushka; Tan, Siok; Pers, Yves-Marie; Vazquez-Alvarez, Raquel; Peñacoba-Maestre, Delia; Baker, Graham; Pilotto, Alberto; Zora, Sabrina; Raat, Hein; Hevia-Fernandez, Jose Ramón

VERSION 1 - REVIEW

REVIEWER	Cesar I Fernandez Lazaro Department of Preventive Medicine and Public Health, School of Medicine, University of Navarra, Pamplona, Spain
REVIEW RETURNED	19-Aug-2019

GENERAL COMMENTS	General Comments: I believe this project has a lot of potential and I am really excited to see the results. I also believe that the project covers a very important issue relevant to public health, chronic conditions and vulnerable populations. My main concerns are primarily providing additional explanations in various sections of the paper. English needs improvement. Specific Comments: Introduction As the EFFICHRONIC project focuses on socio-economic vulnerable populations, the authors should include information and examples of how vulnerable populations, particularly low-income families, are most at risk of developing chronic diseases for several reasons, including greater exposure to risks and decreased access to health services. For example, vulnerable populations are generally less likely to receive preventive care, diagnostic services, and the ongoing follow-up care required to manage chronic conditions. In America individuals living in poverty are much more likely to have poor health and less likely to have access to healthcare. Some data and examples will improve the introduction and rationale of the study. Moreover, authors may want to explain how chronic diseases and vulnerable populations are interconnected in a vicious cycle. Methods and analysis - If at least 500 individuals will be included per region (page 6, line 35 & page 9, line 43), then the minimum sample size will be 2,500 participants. I suggest to modify the study population of 2,500
--

participants to minimum sample size will be 2,500. Here my other concern, if the authors expect a 20% drop out (page 9, line 14), then the minimum sample size for each country has to be 625 participants (20% of 625= 125 then 625-125=500 participants per country in order to meet the authors' proposal). In that case, the minimum sample size has to be 3,125 participants. I suggest to clarify these numbers.

- The authors explained that target population involves "vulnerable population with a chronic disease". I was wondering if the authors want to say "at least one chronic condition" or just "one chronic condition". Since multimorbidity is becoming more prevalent in the overall population (around 80 million Europeans suffer from multimorbidity, whereas in the United States four in ten adults suffer from multimorbidity), the inclusion criteria should not be restricted to individuals with one chronic condition. Moreover, people with multiple chronic conditions are more likely to have more hospitalizations, higher medical care costs, higher number of prescriptions, and lower quality of life (among others negative outcomes) than individuals with one chronic condition. I think the project may be more successful with the inclusion of people with more than one chronic condition. Lastly, the authors stated that the EFFICHRONIC is based on the CDSMP. I am wondering why the EFFICHRONIC does not include individuals with more than one chronic condition if the CDSMP includes "any adults living with one or more chronic conditions".

- My concerns about the inclusion criteria:

1) How do the authors determine if an individual has difficulties to make ends meet or not having means of transportation, internet access or social support? According to the reference #19, vulnerability is based on 5 dimensions (economic, social, familiar, personal, and living). In that reference, economics is defined as income of 500 euros or less. This limit is specific, however, individual with difficulties to make ends is not. Another concern is that the 500 euro-limit will be different for the rest of the European countries of the study.

2) According to #19 reference there is a scale to determine the level of global vulnerability. Do the authors use this scale in the project? I think that using this scale could help to be more specific with inclusion criteria.

3) Authors stated that participants must "have their basic housing needs met and possess an adequate knowledge of the local language". However, and according to #19 reference, lack of housing and do not speak local language are vulnerability criteria (housing and social dimensions respectively). With these exclusion criteria, the design of the study may be susceptible to selection bias.

- What is the auto-administrable questionnaire that will compare the different outcomes at baseline (T0, before the CDSMP) and at 6 months (T1)? Did the authors validate the questionnaire? It would be nice to see it.

Minor comments/suggestions:

- Line 41, page 5: "EU countries". Abbreviations should be defined at the first occurrence.

	- Sample size: I think this section should follow the study population section. The authors mentioned 2,500 participants for first time in the sample study population section (page 77, line 55) and then they justify this number at the end of page 9 (in the sample size section). It would be more appropriate to see how authors reach the proposed sample size after the study population section. - English: The authors need to go through and improve the flow of the manuscript to ensure that the paper reads well and there are no grammatical errors.
--	---

VERSION 1 – AUTHOR RESPONSE

Introduction

The introduction was written in a more clear way and the relation between chronic diseases and vulnerability explained.

Methods and analysis

The sample size has been explained better. The final analysis will be done on 2,000 participants. Assuming a 20% drop-out rate, minimum 2,500 people have to be enrolled in the intervention initially.

The study addresses vulnerable people with one or more chronic conditions and socially isolated caregivers. The inclusion criteria are now better defined in the manuscript.

Inclusion criteria

1) Vulnerability will be measured by means of the Gijón scale, which includes economic, social, familial and housing dimensions. The scale on income is adapted in each country according to the corresponding minimum wages.

2) We do not use the vulnerability scale of the mentioned reference (#19) but a tool specifically designed for the project: the Selfy-MPI. The tool is described in the text and references are provided.

3) We are aware that not to include people without sufficient knowledge of the language is a limitation of the project, and it is now reflected in the summary part of the manuscript. However, to participate in this kind of intervention, it is essential to understand and speak the local language as the sessions are very interactive. Regarding the housing needs, we need to be able to follow-up on the participants at six months and furthermore, people without a home might need other, more urgent interventions before enrolling in a CDSMP intervention.

4) The questionnaire is compiled of several validated scales. The development of the same is described in a paper already published. The reference is added to the manuscript.

All minor comments and suggestions have been considered and general improvements have been made.

VERSION 2 – REVIEW

REVIEWER	Cesar Fernandez Lazaro University of Navarra, Pamplona; Spain
REVIEW RETURNED	15-Oct-2019

GENERAL COMMENTS	General Comments: Authors effectively addressed reviewer's concerns to create a revised manuscript that is more consistent and better explained than the previous version. My only concern is the English language and the flow of the manuscript. I suggest going through the manuscript with an editor, or somebody who is not a part of the study team, to help with final grammar change sand ensure that the manuscript reads well.
--